# Bruton’s Kinase Inhibitors for the Treatment of Immunological Diseases: Current Status and Perspectives

**DOI:** 10.3390/jcm11102807

**Published:** 2022-05-16

**Authors:** Ewa Robak, Tadeusz Robak

**Affiliations:** 1Department of Dermatology, Medical University of Lodz, 90-647 Lodz, Poland; ewarobak@onet.eu; 2Department of Hematology, Medical University of Lodz, 93-510 Lodz, Poland; 3Department of General Hematology, Copernicus Memorial Hospital, 93-510 Lodz, Poland

**Keywords:** AIHA, atopic dermatitis, BTK inhibitor, chronic spontaneous urticaria, IgG4-related disease, ITP, multiple sclerosis, pemphigus vulgaris, rheumatoid arthritis, systemic lupus erythematosus

## Abstract

The use of Bruton’s tyrosine kinase (BTK) inhibitors has changed the management of patients with B-cell lymphoid malignancies. BTK is an important molecule that interconnects B-cell antigen receptor (BCR) signaling. BTK inhibitors (BTKis) are classified into three categories, namely covalent irreversible inhibitors, covalent reversible inhibitors, and non-covalent reversible inhibitors. Ibrutinib is the first covalent, irreversible BTK inhibitor approved in 2013 as a breakthrough therapy for chronic lymphocytic leukemia patients. Subsequently, two other covalent, irreversible, second-generation BTKis, acalabrutinib and zanubrutinib, have been developed for lymphoid malignancies to reduce the ibrutinib-mediated adverse effects. More recently, irreversible and reversible BTKis have been under development for immune-mediated diseases, including autoimmune hemolytic anemia, immune thrombocytopenia, multiple sclerosis, pemphigus vulgaris, atopic dermatitis, rheumatoid arthritis, systemic lupus erythematosus, Sjögren’s disease, and chronic spontaneous urticaria, among others. This review article summarizes the preclinical and clinical evidence supporting the role of BTKis in various autoimmune, allergic, and inflammatory conditions.

## 1. Introduction

In 1952, Dr. Ogden Bruton first reported a case of an 8-year-old boy who presented with recurrent bacterial sepsis, osteomyelitis, and otitis, with the lack of a gamma globulin fraction and failure to produce antibodies [1]. The disease was then described as X- linked agammaglobulinemia (XLA), manifested by decreased serum immunoglobulin and a markedly decreased B-cell number in the peripheral blood. Subsequently, mutations of the gene coding tyrosine kinase, named Bruton’s tyrosine kinase (BTK), were identified [2,3]. Bruton’s tyrosine kinase is a member of the TEC kinase family of nonreceptor tyrosine kinases and is expressed in all hematopoietic cells except T cells, terminally differentiated plasma cells, and natural killer cells [4].

BTK contributes significantly to the proliferation and differentiation of B cells and plays a major role in both the innate and the adaptive immune responses and cytokine production. BTK inhibitors (BTKis) suppress B-cell receptor and myeloid fragment crystallizable receptor mediated signaling, thus inhibiting B-cell activation, antibody class-switching, expansion, and cytokine production. BTKis can affect autoimmune diseases involving B cells and non-B cells through B-cell receptor, Fc receptor, and RANK receptor signaling. Therefore, BTK is currently being investigated as a promising target for the treatment of immunological disorders in addition to B-cell hematological malignancies [5,6]. BTK inhibitors have been investigated in various autoimmune disorders, including autoimmune hemolytic anemia (AIHA), immune thrombocytopenia (ITP), multiple sclerosis (MS), atopic dermatitis (AD), rheumatoid arthritis (RA), systemic lupus erythematosus (SLE), Sjögren’s disease (SD), multiple sclerosis (MS), and pemphigus vulgaris (PV). In addition, BTKis are under investigation for use in allergic and inflammatory diseases including chronic spontaneous urticaria (CSU), asthma, and graft versus host disease (GVHD) [7]. This article summarizes the recent data obtained from preclinical studies and clinical trials of BTKis in various immune-mediated diseases, and this introduction should briefly place the study in a broad context and highlight why it is important.

## 2. The Role of Bruton’s Tyrosine Kinase in the Immune System

Bruton’s tyrosine kinase plays a critical role in the B-cell receptor (BCR) signaling pathway. BTK influences the production of messenger molecules, which can abnormally activate the BCR signaling pathway and transform B cells into self-reactive B cells responsible for autoimmune diseases [8,9,10]. The increased activity of BTK has been observed in several autoimmune disorders, including SLE and RA [11,12]. The signaling potential of BTK, and its targetable nature, have been found to be immensely valuable for a wide range of clinical applications. The development of novel, more specific, and less toxic BTKis could potentially revolutionize the treatment of a significant number of diseases with yet unmet treatment needs [10].

## 3. Characteristics of BTK Inhibitors

The majority of BTKis target the ATP-binding site. These agents are classified as reversible or irreversible BTKis according to their binding mode. In addition, reversible BTKis are divided into two categories, namely covalent reversible inhibitors and non-covalent reversible inhibitors [13,14].

BTKis have shown remarkable efficacy in the treatment of B-cell malignancies, including chronic lymphocytic leukemia (CLL), mantle cell lymphoma (MCL), Waldenstrom macroglobulinemia (MW), and various other B-cell lymphomas. In addition, they have also been investigated in patients with immune disorders, including autoimmune conditions, allergic diseases, and inflammatory disorders. However, long term data are not yet available.

The recent discoveries and significance of BTK inhibitors in the treatment of various immune diseases are summarized in this review.

### 3.1. Irreversible BTK Inhibitors

Covalent, irreversible BTKis bind cysteine 481 (C481) in the ATP binding pocket of BTK, thus inhibiting the phosphorylation of downstream kinases in the BCR signaling pathway and blocking B-cell activation [13]. The C481 in BTK forms a covalent bond with the inhibitor by acting as a nucleophile [15]. Three irreversible covalent BTKis, viz. ibrutinib, acalabrutinib, and zanubrutinib, have been approved for the treatment of lymphoid malignancies [6]. In addition, ibrutinib has been approved for the treatment of chronic GVHD. Of the three, ibrutinib is the most potent BTK inhibitor, followed by zanubrutinib and acalabrutinib, while acalabrutinib is the most selective, followed by zanubrutinib and ibrutinib [16,17]. Recently, several irreversible BTKis have been developed and are under clinical investigation (Table 1) [6,18]. These drugs inhibit BTK activity by irreversible covalent binding with high affinity to the same cysteine 481 in BTK as ibrutinib. Some of the newer BTKis are more selective than currently approved drugs. In addition, the difference in pharmacodynamics and pharmacokinetics between BTK inhibitors may influence their efficacy and safety in clinical use.

#### 3.1.1. Ibrutinib

Ibrutinib (PCYC-1102, Imbruvica^®^, Pharmacyclics/Janssen, Sunnyvale, CA, USA) was discovered in 2007 as an irreversible BTKi with an IC50 for Cys481 binding of 0.5 nM [51]. However, ibrutinib has off-target activity and can block other kinases, including EGFR, ErbB2, ITK, and TEC [15]. This off-target activity is responsible for an increased risk of bleeding, with an inhibition of collagen-induced platelet aggregation and platelet adhesion via the inhibition of intracellular molecules involved in platelet signaling [52]. In addition, ibrutinib affects PI3K-AKT signaling, which is responsible for atrial fibrillation complications in ibrutinib-treated patients [19]. Ibrutinib was approved in 2013 for the treatment of mantle cell lymphoma (MCL) and in 2014 for chronic lymphocytic leukemia (CLL)/small lymphocytic lymphoma (SLL). Subsequently, ibrutinib was approved for Waldenström’s macroglobulinemia (WM), GVHD, and marginal zone lymphoma (MZL). Phase 2 trials in AIHA are ongoing (Table 1).

#### 3.1.2. Acalabrutinib

Acalabrutinib (ACP-196, Calquence^®^, Acerta Pharma, AstraZeneca, Cambridge, UK) is a second-generation oral, highly selective, covalent BTKi approved for the treatment of CLL/SLL and MCL [17]. Acalabrutinib covalently binds to C481 in BTK with an IC50 of 3 nM [20]. The drug demonstrates less off-target binding and a much higher specificity for BTK than ibrutinib. In contrast to ibrutinib, acalabrutinib does not inhibit the TEC-family kinases (Itk and Txk), ERBB2, and Src-kinases (Src, Lyn, Fyn, Yes and Lck) [21]; however, it inhibits the epidermal growth factor receptor (EGFR), which can be associated with a rash and severe diarrhea [22]. In addition, acalabrutinib has a shorter half-life than ibrutinib and is given twice daily [23]. In a phase 3 trial (ELEVATE RR), acalabrutinib was better tolerated and had similar efficacy to ibrutinib in patients with previously-treated CLL when compared with ibrutinib [24]. In particular, cardiovascular events were less common in the acalabrutinib group compared to ibrutinib. Acalabrutinib was approved in 2017 for the treatment of patients with MCL who have received at least one prior therapy, and in 2019 for the treatment of CLL/SLL. It is also under investigation in phase 2 trials in RA, AIHA, and GVHD (Table 1). 

#### 3.1.3. Zanubrutinib

Zanubrutinib (BGB-3111, Brukinsa^®^, BeiGene Beijing, China) is a second-generation irreversible BTKi developed by the Baekje Shenzhou Company of China for the treatment of B-cell lymphoid malignancies [25]. Zanubrutinib demonstrated potent activity and more selectivity against BTK over other TEC, EGFR, and Src families. In addition, compared to ibrutinib, zanubrutinib induced more selective BTK inhibition, with more complete and sustained BTK occupancy, as well as improved oral absorption and better target occupancy. Recent clinical trials have found it to demonstrate excellent efficacy and good tolerability in patients with CLL and WM. In previously treated CLL patients, zanubrutinib demonstrated better efficacy and safety than ibrutinib (ALPINE study) [26]. Currently, zanubrutinib is under investigation in phase 2 trials in patients with immune disorders including ITP, AS (antiphospholipid syndrome), IgG4-RD, and active proliferative lupus nephritis (Table 1).

#### 3.1.4. Spebrutinib

Spebrutinib (CC-292, AVL-292, Avila Therapeutics/Celgene, Summit, NY, USA) is an oral, irreversible, covalent BTKi (Cys481 IC50 < 0.5 nM) that binds covalently with high affinity to the BTK ATP binding site (Cys481 IC50 < 0.5 nM) [27,28]. Spebrutinib has been found to inhibit B-cell proliferation and reduced both lymphoid and myeloid cytokine production and degranulation in vitro, and to influence osteoclastogenesis [28]. Spebrutinib also reduces disease symptoms in experimental RA in mice and in humans [27]. In a first-in-human study performed in healthy volunteers, spebrutinib led to near-complete BTK occupancy for eight to 24 h. Spebrutinib is currently under investigation in a phase 2a, randomized, placebo-controlled study in patients with active RA (Table 1). In a clinical trial with female RA patients on stable methotrexate (MTX), the numbers of naive B cells in circulation and transitional B cells were found to decrease, as was the collagen-degradation product CTX-1, indicating reduced osteoclastogenesis [28].

#### 3.1.5. Evobrutinib

Evobrutinib (A18, M2951, Merck, Readington Township, NJ, USA) is a highly selective, irreversible, covalent BTKi inhibitor (Cys481 IC50 = 9 nM) with high selectivity in inhibiting both BCR and Fc receptor signaling [29,53]. Evobrutinib inhibits B-cell activation and differentiation, as well as M1 macrophage polarization. It also has greater BTK selectivity than ibrutinib. In mouse models of RA and SLE, evobrutinib reduced the disease severity and histological damage [29,54]. In the SLE model, evobrutinib inhibited B-cell activation, reduced autoantibody production, and normalized B- and T-cell subsets. In a phase 1, double-blind, dose-escalation study in healthy participants, evobrutinib showed linear and time-independent PK which induced long-lasting BTK inhibition. The drug was well-tolerated and showed no prolongation of the QT/QTc interval [55]. In this study, full BTK occupancy was achieved with 25 mg of evobrutinib after multiple daily dosing. Treatment-emergent adverse events (AEs) were observed in 25% of volunteers after single dosing and in 48.1% after multiple dosing; however, these tended to be mild in most participants without any dose dependency.

Evobrutinib is one of the first BTK inhibitors to be studied in MS. It has recently completed its phase 2 trial in MS patients [30]. Clinical trials of evobrutinibin in relapsing MS (RMS) and SLE are ongoing (Table 1).

#### 3.1.6. Remibrutinib

Remibrutinib (LOU064, Novartis, Cambridge, MA, USA) is a highly selective, irreversible covalent BTKi with Cys481 IC50 1.3 nM [31]. Remibrutinib very potently inhibits BTK and TEC in vitro and inhibits BTK-dependent platelet activation [32]. In healthy volunteers, single doses of 30 mg remibrutinib or higher resulted in a greater than 95% blood BTK occupancy for at least 24 h. With multiple ascending doses greater than or equal to 10 mg q.d., remibrutinib reached near complete blood BTK occupancy at day 12 [33]. The drug was developed for the treatment of autoimmune disorders, including CSU. In a phase 1 study, remibrutinib showed a favorable safety profile in adult healthy subjects with or without asymptomatic atopic diathesis (NCT03918980). A phase 3 clinical trial in CSU, inadequately controlled by H1 antihistamine (Remix-1), and a phase 2 study in patients with moderate to severe SD (LOUiSSe) are ongoing (Table 1).

#### 3.1.7. Tirabrutinib

Tirabrutinib (Velexbru^®^, ONO/GS-4059, Ono Pharmaceutical, Osaka, Japan) is another very potent and specific covalent, irreversible BTKi that demonstrates greater selectivity than ibrutinib (Cys481 IC50 2 nM) [34]. This agent was developed for the treatment of autoimmune disorders and hematological malignancies. Tirabrutinib has demonstrated significant cytotoxic activity in several types of B-cell neoplasms in vitro and in vivo in mouse models [35]. In mice treated with tirabrutinib, the suppression of osteoclastic bone resorption was observed together with the inhibition of serum TRAPCP5b and urinary CTX-1 [36]. It is currently approved for the treatment of central nervous system (CNS) lymphoma, WM, and CLL [37]. A phase 2 clinical trial in SD is ongoing (NCT03100942).

#### 3.1.8. Elsubrutinib

Elsubrutinib (ABBV-105, Abbvie, Chicago, IL, USA) is an irreversible, non-covalent, highly selective, and potent BTKi with the IC50 of Cys481 binding being 180 nM [38]. The drug inhibits a histamine release from IgE-stimulated basophils and an IL-6 release from IgG-stimulated monocytes. Elsubrutinib also inhibits TNF-release from CpG-DNA stimulated PBMCs [38]. In a preclinical study, the BTK occupancy correlated with the in vivo efficacy. Phase 2 trials to investigate the safety and efficacy of elsubrutinib in RA (NCT03682705) and SLE (NCT03978520) have been initiated.

#### 3.1.9. Tolebrutinib

Tolebrutinib (SAR 442168, PRN 2246, Sanofi/Principia Biopharma, San Francisco, CA, USA) is an investigational irreversible BTK inhibitor. It covalently binds to a specific conserved cysteine present in only 11 kinases, with potential immunomodulatory and anti-inflammatory activities [39,40]. Tolebrutinib can cross the blood–brain barrier and inhibit the activity of BTK in the central nervous system and has potential efficacy in MS. In the first-in-human randomized, double-blind, placebo-controlled study, tolebrutinib was well-tolerated with only mild treatment-related adverse events. Phase 2 and 3 studies against MS and myasthenia gravis (MG) are ongoing (Table 1).

#### 3.1.10. Branebrutinib

Branebrutinib (BMS-986195, Bristol Myers Squib, New York, NY, USA) is a potent, highly selective, oral BTKi that covalently modifies a cysteine residue in the active site of BTK [42]. Branebrutinib has demonstrated 5000-fold higher selectivity for BTK over 240 other kinases. In murine models of collagen-induced arthritis (CIA), branebrutinib reduced clinically evident joint damage and inhibited loss of bone mineral density. Branebrutinib is better tolerated than currently approved, less-selective BTK inhibitors. A randomized phase 1, placebo-controlled trial in healthy participants found BTK to demonstrate a rapid and high occupancy without notable safety findings [43]. Currently, branebrutinib is under evaluation as monotherapy in clinical trials in patients with moderate to severe psoriasis, active SLE and SD, AD, and RA (Table 1).

#### 3.1.11. Orelabrutinib

Orelabrutinib (ICP-022, Biogen/Innocare Pharma, Beijing, China) is another highly selective irreversible, covalent BTKi (Cys481 IC50 = 1.6 nM) under investigation for the treatment of B-cell malignancies and autoimmune diseases [41]. In a KINOME scan assay conducted in parallel against numerous kinases at a drug concentration of 1 μM, orelabrutinib was found to be more selective than ibrutinib. In this study, BTK was the only kinase targeted by orelabrutinib (with >90% inhibition). In 2020, orelabrutinib was approved in China for the treatment of patients with MCL and CLL who had previously received at least one line of anticancer therapy [56]. It is currently investigated in clinical trials for lymphoid malignancies and autoimmune disorders, including ITP (NCT05020288, NCT05124028), SLE and SD (NCT04305197, NCT04186871), and relapsing-remitting MS (NCT04711148).

#### 3.1.12. Poseltinib

Poseltinib (HM71224, LY3337641, Hanmi Pharmaceutical, Eli Lilly, Indianapolis, IN, USA) is an experimental, selective irreversible, non-covalent BTKi with potential anti-inflammatory activity which was developed for the treatment of RA and SLE [44,45]. The drug is relatively selective for a conversed cysteine shared by only ten kinases within the kinome. In a preclinical study, poseltinib effectively inhibited the production of tumor necrosis factor (TNF)-α, interleukin (IL)-6, and IL-1β by human monocytes, as well as osteoclast formation by human monocytes. It also reduced the signs and symptoms of arthritis and prevented joint destruction in a dose-dependent manner in a murine and rat CIA model [44]. A clinical phase 1 trial for healthy volunteers, showed a dose-dependent and persistent BTK occupancy in the peripheral blood mononuclear cells of all patients receiving poseltinib [46]. A first-in-human healthy volunteer study confirmed poseltinib as a potential BTK inhibitor for the treatment of autoimmune diseases [46].

#### 3.1.13. SHR1459 

SHR1459 (TG 1701, EBI-1459; Reistone Biopharma, Jiangsu Hengrui Medicine Co, Lianyungang, China) is a covalently bound and irreversible BTKi currently under clinical development. This agent exhibits superior BTK selectivity compared to ibrutinib in in vitro kinase screening [47,48]. SHR1459 is currently under clinical development in phase 2 trials in membranous glomerulonephritis and neuromyelitis optica (Table 1).

#### 3.1.14. TAS5315

TAS5315 (SAT0056, Taiho Pharmaceutical Co, Tokyo, Japan) is a novel covalent, irreversible, highly selective BTKi with significant efficacy in a mouse CIA model for RA. TAS5315 treatment resulted in a dose-dependent reduction of the clinical score in a mouse CIA model compared with vehicle-treated mice. When applied at 0.4 mg/kg, TAS5315 completely ameliorated arthritic symptoms on day 14 [49]. In a histopathological analysis, the mice treated with TAS5315 demonstrated a marked reduction in the severity of inflammation, pannus, cartilage destruction, and bone destruction in a dose-dependent manner. TAS5315 inhibited RANKL-dependent osteoclast activation and osteoclast activation induced by inflammatory cytokines [50]. A phase 2 study comparing TAS5315 with placebo in participants with RA is ongoing (Table 1).

#### 3.1.15. AC0058 

AC0058 (ACEA Biosciences, San Diego, CA, USA) is a covalent, irreversible BTKi. In preclinical studies, AC0058 inhibited B-cell activation and inflammatory cytokine production in monocytes [5]. The drug is in development for the treatment of B cell-related autoimmune diseases, including RA and SLE. In a double-blind, placebo-controlled phase 1 trial, AC0058 was found to be safe and well-tolerated (NCT02847325). A phase 1b double-blind, randomized, placebo-controlled study of the safety, pharmacokinetics, and pharmacodynamics of AC0058 in patients with SLE is ongoing (Table 1).

### 3.2. Reversible BTK Inhibitors

Reversible BTKis can be divided into covalent reversible inhibitors and non-covalent reversible inhibitors (Table 2), with the two acting through different mechanisms: the non-covalent inhibitors do not bind to the C481 site on BTK. They hence provide an effective alternative to patients with B-cell malignancies who have developed resistance following prior therapy with covalent BTKis [57,58].

Some reversible BTKis (fenebrutinib, nemtabrutinib, rilzabrutinib, pirtobrutinib, and PRN473) have entered early clinical trials [13,57]. They were found to required more frequent and sustained dosing but have less off-target effects on other kinases. Reversible inhibitors seem to be more effective in the treatment of autoimmune diseases such as RA, different types of MS, GVHD, and SLE [18]. However, while clinical trials with these drugs are promising, they are less advanced than those with irreversible BTK inhibitors. These agents have shown encouraging results in preclinical studies and some in early clinical trials.

#### 3.2.1. Fenebrutinib

Fenebrutinib (GDC-0853, Roche/Chugai Pharmaceutical, Tokyo, Japan) is a selective, reversible, non-covalent BTKi that has a distinct kinase selectivity profile, with strong inhibitory activity against several key oncogenic drivers from the TEC, Trk, and Src family kinases [59]. Fenebrutinib forms hydrogen bonds with K430, M477, and D539. The agent blocks the IgE-mediated histamine release from the mast cells in vitro, and it was found to inhibit IgE-mediated basophil activation in healthy individuals in a phase 1 study [60]. In RA patients with an inadequate response to methotrexate, fenebrutinib caused dose-dependent reductions in the rheumatoid factor with comparable efficacy to adalimumab [61]. A similar response was observed in a phase 2 study performed in SLE patients, where fenebrutinib caused dose-dependent reductions in anti-double-stranded DNA autoantibodies [62]. However, in moderate to severe SLE patients, no improvement was observed in the symptoms (Table 2) [63]. In a placebo-controlled, randomized phase 2 trial, fenebrutinib decreased the disease activity in patients with antihistamine-refractory CSU [64], and the drug is currently undergoing clinical development for autoimmune disorders such as RA and SLE (Table 2).

#### 3.2.2. Nemtabrutinib

Nemtabrutinib (MK1026, ARQ 531, ArQule, Inc., Woburn, MA, USA) is a potent, reversible inhibitor of both a wild-type and ibrutinib-resistant C481S-mutant BTK [69], which binds to BTK by forming hydrogen bonds with E475 and Y476. It has a distinct kinase selectivity profile, with strong inhibitory activity against several key oncogenic drivers from the TEC, Trk, and Src family kinases. In addition to inhibiting BTK, nemtabrutinib also inhibits other kinases involved in BCR signaling including the SRC family kinase LYN and the immediate upstream kinase of ERK and MEK1 [70]. Studies evaluating the safety, tolerability, and pharmacokinetics (PK) of nemtabrutinib have been initiated in patients with hematologic malignancies and mild to moderate persistent asthma (Table 2).

#### 3.2.3. Pirtobrutinib

Pirtobrutinib (LOXO-305, Loxo Oncology, Stamford, CT, USA) is a reversible, non-covalent, next-generation BTKi which blocks the ATP binding site of BTK by noncovalent, non-C481-dependent binding [74]. Pirtobrutinib is under evaluation in patients with B-cell malignancies. However, clinical trials in immune-mediated diseases have not been initiated yet.

#### 3.2.4. Rilzabrutinib

Rilzabrutinib (PRN1008, Principia Biopharma/Sanofi, Paris, France) is another reversible, covalent BTKi with a Cys481 IC50 of 1.3 nM [65]. The drug has shown high affinity and selectivity for the BTK, together with a long duration of action due to prolonged, but reversible, target occupancy [66,67]. In contrast to ibrutinib, rilzabrutinib does not interfere with normal platelet aggregation. The agent has demonstrated anti-inflammatory effects, the neutralization of pathogenic autoantibodies, and the blockade of new autoantibody production. In a preclinical study, rilzabrutinib inhibited the activation and inflammatory activities of B cells, macrophages, basophils, mast cells, and neutrophils in human and rodent assay systems [68]. In addition, administration improved joint pathology in a rat CIA model and reduced autoantibody mediated FcγR signaling in vitro and in vivo.

In contrast to ibrutinib, preclinical studies showed simultaneous rapid anti-inflammatory effects, as well as the neutralization and prevention of autoantibody signaling without any effect on collagen-induced platelet aggregation [65]. The preclinical and clinical data suggest that rilzabrutinib has the potential to treat a wide range of immune-mediated diseases. A phase 1 study in healthy volunteers and patients with immune-mediated diseases found rilzabrutinib to have a promising safety profile [66,67]. In earlier studies, rilzabrutinib did not impact platelet aggregation in blood samples from either normal healthy volunteers or ITP patients [68]. It also inhibits platelet destruction, mainly via the inhibition of autoantibody/FcγR signaling in splenic macrophages [68]. This observation is important in ITP patients because of the associated reduction of the risk of bleeding and bruising typically associated with most BTK inhibitors.

Rilzabrutinib is currently under investigation for ITP in phase 2 and phase 3 studies (Table 2). It is also the subject of phase 2 studies for AIHA, IgG4-related disease (IgG4RD), PV, asthma, CSU, and AD (Table 2).

#### 3.2.5. PRN473

PRN473 (SAR 444727, Principia/Sanofi, Paris, France) is a topical, reversible covalent BTKi designed for immune-mediated diseases that could benefit from a localized application to the skin with low to no systemic exposure [71]. PRN473 inhibits the IgE (FcεR)-mediated activation of mast cells and basophils, IgG (FcγR)-mediated activation of monocytes, and neutrophil migration. When administered orally, PRN473 was found to be effective and well tolerated in the treatment of canine pemphigus foliaceus [72]. A phase 2a study of the safety, tolerability, and pharmacokinetics of topically administered PRN473 in patients with mild to moderate atopic dermatitis is ongoing (Table 2).

#### 3.2.6. BMS-986142

BMS-986142 (Bristol-Myers Squibb, New York, NY, USA) is a reversible covalent BTKi found to reduce FcR-mediated cytokine production and BCR-induced cytokine production by normal B cells in vitro [73]. In animal models of RA, BMS-986142 reduced joint inflammation and destruction. However, in a clinical trial with active RA patients, treatment with BMS-986142 and MTX had no advantage over placebo and MTX (Table 2) [63].

#### 3.2.7. TAS5315

TAS5315 (Taiho Pharmaceutical Co, Tokyo, Japan) is a novel, irreversible, covalent, highly selective BTKi with a Cys481 IC50 value of < 0.15 nM [75]. Yoshiga et al. evaluated the effect of TAS5315 on bone erosion and the expression of inflammatory factors from macrophages and osteoclasts by time-dependent micro-CT analysis in mouse CIA [76]. TAS5315 was found to selectively inhibit BTK with less off-target activity than other kinases. In addition, it was more active against osteoclastogenesis and bone resorbing activity in osteoclasts compared with CC-292. In a CIA model, TAS5315 significantly reduced paw swelling, and treatment was associated with the repair of bone erosion and improved bone mineral density from joint destruction. These results indicate that TAS5315 is a very potent agent for use against joint bone damage and inflammation. It also improved bone erosion in a murine model for RA, suggesting that it may merit clinical studies.

#### 3.2.8. GDC-0834

GDC-0834 (Genentech, San Francisco, CA, USA) is a highly potent, selective, reversible adenosine triphosphate (ATP)-competitive BTK inhibitor developed as a potential drug for treating RA and other immune and inflammatory diseases [77]. This agent inhibited BTK with an in vitro IC_50_ of 5.9 nM in a biochemical assay and 6.4 nM in a cellular assay. It was found to have an IC_50_ of 1.1 in mice and 5.6 microM in rats in vivo. GDC-0834 inhibits Toll-like receptor 4 and tumor necrosis factor-α (TNF-α) in monocytes, which may also contribute to pathogenesis in RA [78]. In CIA rat models, treatment with oral GDC-0834 demonstrated a robust anti-arthritis effect with a dose-dependent decrease of ankle swelling and a reduction of morphologic changes. GDC-0834 has recently been in phase 1 clinical trials for the treatment of RA but the study has been discontinued.

#### 3.2.9. CGI-1746

CGI-1746 (CGI/Genetech, San Francisco, CA, USA) is a reversible, non-covalent ATP-competitive BTKi with anti-inflammatory and anti-arthritic activity in experimental mouse models. This agent reduced cytokine and autoantibody levels in the joints in a mouse arthritis model induced by anti-collagen II antibodies [79]. CGI-1746 inhibited BCR-mediated B-cell proliferation and suppressed FcγRIII-induced TNFα, IL-1β, and IL-6 production in macrophages in cellular assays [80]. However, clinical trials have not yet been initiated.

#### 3.2.10. BIIB068

BIIB068 (BIIB068, Biogen, Cambridge, MA, USA) is a reversible BTKi showing over 400-fold selective inhibition for BTK compared to other kinases [81]. It has the potential for autoimmune diseases. A Phase 1, Single-Ascending-Dose, Safety, Tolerability, Pharmacokinetic (PK), and Pharmacodynamic (PD) study of BIIB068 in healthy participants is ongoing (Table 2).

#### 3.2.11. RN-486

RN-486 (Roche, Basel, Switzerland), is a selective and reversible BTKi that potently and competitively binds to the kinase. Preclinical studies indicate that the drug has therapeutic activity in RA in rodents and RA synovial tissue explants [82,83]. In addition, in mouse models of SLE, RN486 decreased SLE symptoms by inhibiting B-cell activation and reducing the secretion of IgG anti-double-stranded DNA (anti-dsDNA) [84]. However, clinical trials with RN-486 have not yet been registered.

## 4. Autoimmune Hemolytic Anemia

Autoimmune hemolytic anemia is an autoimmune disease characterized by antibodies directed against red blood cells (RBCs). AIHA is idiopathic (primary) in 50% of patients, and secondary to an underlying condition, including lymphoproliferative disease, infections, immunodeficiency, and other diseases, in the other 50% [85]. AIHA is also divided into the more common (70–80%) warm AIHA and less common (20–30%) cold AIHA, according to the thermal characteristics of the autoantibodies [86]. Humoral, cellular, and innate immunity, including warm polyclonal high-affinity IgG autoantibodies, are believed to play a role in the pathogenesis of warm AIHA, while in cold AIHA, monoclonal IgM is also able to fix complement at low temperatures [87].

In patients with coexisting AIHA and progressive CLL, no progression of CLL was observed following ibrutinib treatment [88,89]. In addition, several recent studies have demonstrated that ibrutinib is an effective drug in patients with AIHA complicated lymphoid malignancies [90,91]. It was also found that AIHA was suppressed during ibrutinib therapy, but worsened when ibrutinib was withdrawn [90,92]. AIHA has been found to develop in patients treated with ibrutinib [93,94]; however, only 1.28% of AIHA were attributed to the use of ibrutinib in one study [89]. Ibrutinib has also been successfully used to treat patients with both AIHA and CLL; after treatment, they discontinued therapy specifically for AIHA [89,95]. Ibrutinib may be also an effective drug in the treatment of patients with cold AIHA, with all 13 patients with cold AIHA in a recent study showing an improvement in hemoglobin levels, including 12 complete responses (CR) and one partial response (PR), after the initiation of ibrutinib [96].

Acalabrutinib, a second-generation selective BTK inhibitor, also reduced concomitant AIHA in patients with relapsed/refractory CLL [97]. This drug is currently under evaluation in patients with AIHA and CLL in a phase 2 study (Table 1). Rilzabrutinib, a reversible BTK inhibitor, is also under investigation in patients with idiopathic warm AIHA in a multicenter, phase 2b Study (Table 2); however, the results are not yet available.

## 5. Immune Thrombocytopenia

Immune thrombocytopenia is an acquired disease characterized by the autoimmune destruction of platelets and megakaryocytes by IgG autoantibodies targeting surface antigens such as glycoprotein (GP) αIIbβ3 (GPIIbIIIA) and GPIb-IX-V [98]. Patients with B-cell malignancies treated with irreversible BTKis such as ibrutinib, acalabrutinib, zanubrutinib, and tirabrutinib often demonstrate mild bleeding [21]; however, no bleeding has been reported in the clinical trials of tolebrutinib, evobrutinib, branebrutinib, BMS-986142, fenebrutinib, and rilzabrutinib.

Ibrutinib administration inhibits platelet aggregation, and this precluded its use during ITP. However, it has shown beneficial effects in autoimmune cytopemias in patients with CLL [95]. Hampel et al. reported that among 193 patients treated with ibrutinib for CLL, 29 demonstrated autoimmune cytopenias prior to the initiation of treatment, eight had ITP, and five had Evans’ syndrome [95]. No progression of autoimmune cytopenias was observed. In addition, 42% of the patients underwent the reduction of treatments specific for autoimmune disorders and 25% discontinued therapy dedicated to autoimmune cytopenia.

Another irreversible BTKi under investigation for primary ITP in a phase 2 clinical trial is zanubrutinib (Table 1). Zanubrutinib is less potent than ibrutinib in inducing the shedding of human and mouse GPIbα and GPIX in vitro and in vivo from the platelet surface. In contrast to ibrutinib, zanubrutinib does not induce the shedding of human and mouse integrin α_IIb_β_3_ in vitro or in vivo [99]. These results can explain a lower risk of bleeding complications in zanubtutinib-treated CLL patients.

Rilzabrutinib is a reversible BTK inhibitor that influences mechanisms driving ITP [65]. In contrast to ibrutinib, rilzabrutinib has no effect on collagen-induced platelet aggregation, nor bruising, nor any fluctuation of the platelet count in healthy people [32,100]. Recently, the preliminary results of a phase 1/2 study of rilzabrutinib in 32 chronic ITP patients were reported [101]. Rilzabrutinib was well-tolerated at all studied doses, and an optimal dose of 400 mg bid was established. Treatment resulted in a rapid and durable platelet response across several subgroups of patients and with extended treatment. Half of the patients achieved a platelet count of ≥30 G/L in the first week of treatment, with 42% achieving ≥50 G/L. In addition, 71% of the patients demonstrated a persistent response lasting several weeks. Rilzabrutinib is currently being compared with placebo in adult and adolescent patients with persistent or chronic ITP in the ongoing phase 3 LUNA3 study, the first randomized, multicentered study evaluating its efficacy and safety, initiated in 2020 (NCT04562766; EudraCT 2020-002063-60). Eligible patients are randomized into receiving oral rilzabrutinib 400 mg bid or placebo for up to 24 weeks, followed by open-label treatment for 28 weeks, and then a four-week safety follow-up. Their findings should deliver data on the role of BTKis in the treatment of ITP [67,102]. Rilzabrutinib is being investigated in a phase 2 and phase 3 trial in patients with ITP (Table 2).

Orelabrutinib is another selective BTK inhibitor of potential value in the treatment of ITP. In a preclinical study, orelabrutinib effectively suppressed the activation and differentiation of B cells in vitro and in vivo, and reduced thrombocytopenia in active ITP murine models [103]. Phase 2 clinical trials in patients with primary, refractory ITP are ongoing (Table 1).

## 6. Rheumatoid Arthritis

Rheumatoid arthritis is a systemic autoimmune disease that causes chronic inflammation of the joints and surrounding tissues, as well as pathological bone resorption due to osteoclast activity, resulting in joint destruction and disability. The disease is characterized by B-cell dysregulation via BCR signaling, and dysregulation of T cells, leading to the production of autoantibodies and inflammatory cytokines involved in the development and progression of RA. This may indicate that BTKis can be useful drugs in the treatment of RA [104].

Several preclinical studies indicate that BTKis may have therapeutic activity in RA. It is believed to act by decreasing the formation of myeloid immune complexes, inhibiting B-cell activation and inflammatory cytokines, and suppressesing osteoclastic bone erosion [28,73]. Most studies investigating BTK inhibitors in RA, based on a collagen induced arthritis (CIA) rodent model, indicate that BTKis inhibit B-cell activation, myeloid immune complexes’ formation, decrease inflammatory cytokine activity, and stimulate osteoclastogenesis [28,73,78,79,105].

Evobrutinib, spebrutinib, and GDC-0834 significantly reduced the development and disease activity in CIA in rodents [27,29]; however, GDC-0834 yielded similar effects to the controls treated with methotrexate [78]. The irreversible tricyclic BTKi SOMCL-17-016 was found to ameliorate arthritis and bone damage in CIA mice more effectively than corresponding doses of ibrutinib and acalabrutinib; it was also found that SOMCL-17-016 demonstrated therapeutic effects in the CIA model, inhibiting the B-cell function and osteoclastogenesis [105].

Several BTKis, such as spebrutinib (CC-292), ABBV-105, evobrutinib, BMS-986142, rilzabrutinib, and acalabrutinib, have been studied in phase 1 trials in recent years. All these trials reported positive results without serious safety concerns. They also achieved sufficient receptor occupancy with satisfactory pharmacokinetic (PK) and pharmacodynamic (PD) endpoints and were found to be effective in animal models [20,28,29,38,106,107]. More recently, irreversible and reversible BTK inhibitors, including acalabrutinib, spebrutinib, evobrutinib, tirabrutinib subrutinib, branebrutinib, poseltinib, TAS5315, and fenebrutinib, are undergoing phase 2 clinical trials (Table 1 and Table 2) [36,104,107,108,109].

Shafer et al. evaluated the pharmacology and mechanism of the action of spebrutinib in patients with active RA based on the American College of Rheumatology ACR20 endpoint [28]. It was found that 41.7% (10/24) of RA patients treated with spebrutinib and 21.7% (5/23) of placebo patients demonstrated ≥ 20% improvement at week four; however, this difference was not statistically significant (*p* = 0.25). Even so, a subgroup analysis indicated potential efficacy in female patients with RA.

Fenebrutinib has shown efficacy in reducing RA activity in patients with an inadequate response to either methotrexate or TNF inhibitors. In a multicenter, randomized, double-blind Phase 2 study (ANDES), fenebrutinib was compared to placebo and adalimumab in patients with an inadequate response to either methotrexate or TNF inhibitors [61]. Patients with RA and an inadequate response to methotrexate received fenebrutinib (50 mg once daily, 150 mg once daily, 200 mg twice daily), 40 mg adalimumab every other week, or placebo. In the second cohort, patients with RA and an inadequate response to tumor necrosis factor inhibitors were treated with fenebrutinib (200 mg twice daily) or placebo. Both groups continued treatment with methotrexate. In cohort 1, those receiving fenebrutinib 50 mg once daily demonstrated a similar ACR50 at week 12 to placebo (15%); however, higher ACR50 scores were observed for those receiving fenebrutinib 150 mg once daily (28%) and 200 mg twice daily (35%) (*p* = 0.017; *p* = 0.0003). In cohort 2, more patients treated with fenebrutinib 200 mg twice daily achieved ACR50 (25%) than those with placebo (12%) (*p* = 0.072). The most common adverse events for fenebrutinib were nausea, headache, anemia, and upper respiratory tract infections.

BMS-986142, another reversible, covalent BTKi, was evaluated in a phase 2 study; the participants included patients with moderate to severe RA with an inadequate response to methotrexate or methotrexate and who had received up to two TNF inhibitors (Table 2). In the phase 2 study, the patients were randomized to either one of the three doses of BMS-986142 or placebo as 1:1:1 randomization for 12 weeks. Although the study has been completed, the results are not available yet.

The results of the phase 2 RA-JUVENATE trial investigating poseltinib in patients with active RA were recently published [109]. The trial included patients with active RA, and with either an inadequate response or loss of response to at least one disease-modifying antirheumatic drug, or who had an intolerance to these agents. No statistically significant difference in the ACR20 response was observed between any dose of poseltinib and placebo at week 12 (*p* > 0.05 for all comparisons) in the interim analysis, and part B of the study was discontinued.

GDC-0834 is a highly selective, reversible BTKi that was developed as a potential drug for RA [77]. It inhibited BTK phosphorylation in a preclinical study in vitro and suppressed arthritis in a rat CIA model in vivo [78]. Moreover, GDC-0834 significantly inhibited joint inflammation, cartilage damage, and bone resorption.

## 7. Systemic Lupus Erythematosus

Systemic lupus erythematosus (SLE) is an autoimmune disease characterized by autoantibody formation, immune complex deposition, multisystem involvement, and tissue damage [110]. Several components of BTK signaling pathways are altered in B cells from patients with SLE, and BTKis are promising for the treatment of SLE. 

BTK inhibitors (PF-06250112, M7583, BI-BTK-1, RN-486, ibrutinib, poseltinib, and evobrutinib) have demonstrated the prevention or amelioration of lupus nephritis and other SLE symptoms in mouse models of SLE [45,111,112,113].

M7583 was investigated in the BXSB-Yaa and pristane-DBA/1 mouse lupus models [114]. In BXSB-Yaa lupus mice, M7583 reduced autoantibody levels, nephritis, and mortality. In the pristane-induced DBA/1 lupus model, this agent suppressed arthritis but did not affect autoantibodies or the IFN gene signature. Another irreversible BTK inhibitor, BI-BTK-1, ameliorated multiple pathological endpoints associated with kidney disease in two distinct murine models of spontaneous lupus nephritis in mouse models of SLE: NZB × NZW F1 (NZB/W) and MRL/lpr [111]. Treatment with BI-BTK-1 significantly protected both strains from the development of proteinuria, decreased anti-DNA titers, and increased survival. However, lupus nephritis or kidney damage is often reduced, despite the persistence of autoantibodies in the serum, especially anti-RNA antibodies [114].

Fenebrutinib was investigated in a phase 2, multicenter, randomized, placebo-controlled study in patients with moderately-to-severely active SLE. The study demonstrated significant reductions in CD19+ B cells and anti-dsDNA autoantibodies, but no significant clinical response was observed [62]. Treatment was well tolerated but the primary end point of the study, SLE Responder Index 4 (SRI-4) response at week 48, was not met. A phase 1 clinical trial with AC0058 in SLE patients is ongoing (Table 1).

## 8. Sjogren’s Disease

Sjogren’s disease (SD) was first described by Swedish physician Henrik Sjögren, who reported the coincidence of profound dry eyes and dry mouth with chronic arthritis in a group of women. Primary SD is a chronic inflammatory rheumatic disease resulting in immune-mediated injury to the lacrimal and salivary glands. Clinical trials have been initiated in SD, with these acting on the autoimmune processes believed to play a role in its pathogenesis. Autoreactive CD19-hBtk mice develop spontaneous SD/SLE-like autoimmune symptoms, including lymphocytic infiltrates in the salivary glands [115]. In addition, patients with active SD had increased BTK levels in circulating B cells, correlating with the numbers of infiltrating T cells in the parotid gland [116].

BMS-986142, branebrutinib, remibrutinib, and tirabrutinib have been tested in SD in several clinical trials. The efficacy of treatment with either lulizumab or BMS-986142 versus placebo was evaluated in a phase 2 study in patients with moderate to severe primary SD. The efficacy was measured by the change from the baseline in the ESSDAI score at week 12 between active treatment arms (lulizumab or BMS-986142) and the placebo arm. However, the study was terminated prematurely due to a lack of therapeutic activity. In addition, treatment with branebrutinib and tirabrutinib did not yield any clinical improvement [9].

A phase 2 randomized double-blind, placebo-controlled multi-center study evaluating the safety and efficacy of multiple remibrutinib doses in patients with moderate to severe SD (LOUiSSe) is ongoing (Table 1) and should be completed in 2024.

## 9. Systemic Sclerosis

Systemic sclerosis (SS) is a connective tissue disease affecting the skin, blood vessels, and internal organs, with key roles in its pathogenesis played by immune dysregulation and autoantibody production [117], vasculopathy, and chronic fibroblast activation. An in vitro study found ibrutinib to reduce the production of the profibrotic hallmark cytokines IL-6 and TNF-α from the effector B-cell population in primary samples from patients with SS. In addition, small doses of ibrutinib (0.1μM) preserved the production of immunoregulatory IL-10 and inhibited hyperactivated, profibrotic effector B cells. It appears that ibrutinib improves the B-cell pathology contributing to SS pathogenesis and progression, and the findings provide support for the initiation of clinical trials of BTK inhibitors in humans.

## 10. Multiple Sclerosis 

Multiple sclerosis is an autoimmune disorder caused by the action of environmental factors in genetically predisposed hosts. The B cells in the brain play a central role in antigen presentation, cytokine secretion and antibody production, and in the pathogenesis of the disease [118]. In response, BTKis can penetrate to the brain and spine, reducing inflammation and neurodegeneration within the central nervous system, and target the microglia of brain-resident B cells; they can also attenuate B-cell:T-cell interactions [119].

Preclinical studies in animal models of MS indicate that BTKis influence meningeal inflammation and cortical demyelination [120,121]. Hence, several irreversible (evobrutinib, tolebrutininb, and orelabrutinib) and reversible BTKis (fenebrutinib and BIIB091) might be effective in the treatment of patients with MS [122]. In a preclinical study, evobrutinib inhibited antigen-presenting B-cells for the development of encephalitogenic T cells in mice and reduced disease severity [123].

In a double-blind, randomized, phase 2 trial, performed in relapsing MS, patients received evobrutinib at a dose of 25 mg once daily, 75 mg once daily, 75 mg twice daily, or open-label dimethyl fumarate (DMF). Another group received a placebo [124]. Patients who received 75 mg of evobrutinib once daily had significantly fewer enhancing lesions during weeks 12 through 24 in comparison with the placebo group. However, no significant improvement was observed among patients treated with evobrutinib at doses of 25 mg once-daily or 75 mg twice-daily, nor was there any improvement in the annualized relapse rate or the disability progression observed at any dose. Subsequently, the phase 3 EVOLUTION trial evaluating the efficacy and safety of evobrutinib, given orally twice daily, versus Teriflunomide (Aubagio^®^), given orally once daily, has been initiated in patients with relapsing MS (RMS) (Table 1).

Tolebrutinib (PRN2246, SAR442168) is a covalent BTKi that crosses the blood–brain barrier and potently inhibits BTK in microglial cells isolated from the central nervous system (CNS) [40]. Tolebrutinib was investigated in a phase 2b, randomized, double-blind, placebo-controlled trial in relapsing-remitting or relapsing secondary progressive MS [125]. Treatment with tolebrutinib for 12 weeks led to a dose-dependent reduction in new gadolinium-enhancing lesions and in new or enlarging T2 lesions. The most common AE during tolebrutinib treatment was a headache. No safety- or treatment-related discontinuations were observed. Phase 3 trials in RMS, primary progressive (PPMS), and secondary progressive (SPMS) forms of MS are ongoing.

Evobrutinib was found to be effective against experimental autoimmune encephalomyelitis, an animal model of brain inflammation [123]. The drug was also evaluated in a phase 2 clinical trial in relapsing-remitting MS and active SPMS (Table 1) [124]. In this trial, the patients received placebo or evobrutinib at a dose of 25 mg once daily, 75 mg once daily, 75 mg twice daily, or open-label dimethyl fumarate (DMF) as a reference. The primary end point was the total (cumulative) number of gadolinium-enhancing lesions identified on T_1_-weighted magnetic resonance imaging at weeks 12, 16, 20, and 24. Treatment with evobrutinib significantly reduced relapse rates and brain lesions at 48 weeks in a dose-dependent manner in relapsing MS compared with the placebo. The mean total number of gadolinium-enhancing lesions during weeks 12 through 24 was 4.06 in the evobrutinib 25-mg group, 1.69 in the evobrutinib 75-mg once-daily group, 1.15 in the evobrutinib 75-mg twice-daily group, 3.85 in the placebo group, and 4.78 in the DMF group. No significant change in the EDSS score from the baseline was observed in any trial group. This reduction was sustained for about two years. The improvement was especially apparent in patients with a more advanced disease [126]. Evobrutinib is currently under investigation in patients with relapsing MS in two identically-designed phase 3 trials: EVOLUTION RMS 1 and EVOLUTION RMS 2 (Table 1).

## 11. Pemphigus Vulgaris

Pemphigus vulgaris is a potentially life-threatening autoimmune disease characterized by acantholysis, which results in erosions and blisters of the skin and mucous membrane. In this disease, pathogenic autoantibodies act against intercellular adhesion proteins in the epidermis of the skin and mucous membrane. It is found in two major clinical forms: pemphigus vulgaris and pemphigus foliaceus (PF). Pemphigus vulgaris affects both the skin and mucous membrane when PF affects only the skin. Currently, the standard of care for the treatment of this disease is systemic corticosteroid treatment [127]. BTK inhibitors are attractive drugs for the treatment of pemphigus as they influence the innate immune system.

Early studies performed in canine pemphigus showed that rilzabrutinib treatment was safe and effective clinically. It is also being investigated in PV (Table 2) [65,67,128]. In addition, the efficacy and safety of oral rilzabrutinib in PV was confirmed in a recent multicenter, phase 2 trial, with 52% and 70% of patients showing control of disease activity at four weeks and eight weeks, respectively [67]. However, the phase 3 placebo-controlled trial (PEGASUS) evaluating rilzabrutinib in patients with pemphigus did not show any difference between rilzabrutinib and the placebo with regard to its primary endpoint, i.e., complete remission from week 29 to week 37 (Table 2, Sanofi Press Release (https://www.sanofi.com/en/media-room/press-releases/2021/2021-09-09-07-00-00-2293920, accessed on 9 September 2021.

PRN-473 was active in the treatment of canine PF [128]. The treated dogs showed a reduction in lesions and a modified version of a validated human Pemphigus Disease Activity Index (cPDAI) score during the first two weeks of treatment in four of nine good responses at the end of treatment after a maximum of 20 weeks. In rats, the topical administration of PRN-473 inhibited an IgG-mediated passive Arthus reaction. Similarly, topical PRN473 administration caused a significant dose-dependent inhibition of the passive cutaneous anaphylaxis IgE-mediated reaction in mice. A low systemic accumulation of the drug was observed, if any. These experiments deliver strong support for local activity in innate immune cell responses with minimal systemic exposure.

## 12. Allergic Diseases

BTK inhibitors offer several advantages over other drugs commonly used for treating allergic diseases, including the inhibition of IgE-FcϵRI-mediated activation of both mast cells and basophils. The initial results of a phase 2 trial showed that treatment with ibrutinib therapy suppresses skin test responses and eliminates IgE-mediated basophil activation in adults with peanut or tree nut allergy [129,130]. BTKis may be effective in the treatment of allergic and atopic skin reactions, as well as in IgE-mediated anaphylaxis. Short treatment with ibrutinib transiently reduced skin prick test responses to foods and aeroallergens in allergic patients. These observations indicate that short courses of ibrutinib could be effective in preventing anaphylaxis to foods or drugs. However, BTKis are not likely to be effective in IgE-independent allergic diseases. Preclinical studies and clinical trials suggest that BTK inhibitors can be useful for the treatment of IgE-dependent anaphylaxis, food and drug allergy, asthma, CSU, and other allergic diseases.

### 12.1. Atopic Dermatitis

Atopic dermatitis is a chronic, pruritic inflammatory skin disease typically affecting the face, neck, arms, and legs [131]. In a study of adults with a peanut or tree nut allergy, testing allergic patients before and after the initiation of treatment with ibrutinib showed the elimination of both the aeroallergen skin test and the basophil activation test reactivity, along with IgE-mediated basophil activation [129,132].

Tolebrutinib has also been evaluated against AD. It is a reversible covalent BTKi in a topical formulation designed for the treatment of immune-mediated skin diseases that could benefit from a localized application to the skin. A phase 2a, randomized, double-blind, placebo-controlled trial was designed to evaluate the safety, tolerability, and pharmacokinetics of topical PRN473 in healthy adult participants (Table 1).

### 12.2. Asthma

Asthma is a complex airways disease with a wide spectrum which ranges from eosinophilic (Th2 driven) to mixed granulocytic (Th2/Th17 driven) phenotypes. A particular cause for concern is mixed granulocytic asthma, as it is often unresponsive to corticosteroids. However, Bruton’s tyrosine kinase plays an important role in shaping allergic airway inflammation [133]. Ibrutinib was able to suppress both Th17/Th2 and neutrophilic/eosinophilic inflammation in a mouse model of mixed granulocytic asthma, suggesting that this drug may be an alternative therapeutic option in corticosteroid treatment-resistant asthma.

## 13. Other Immunological Diseases

### 13.1. Chronic Spontaneous Urticaria

Chronic spontaneous urticaria is characterized by itching and hives lasting for more than six weeks, with or without angioedema [134]. BTK inhibitors can be effective for treating both autoallergic and autoimmune CSU, as BTK mediates degranulation in mast cells and participates in MCs and autoantibody production in B cells. The efficacy and safety of remibrutinib was tested in a randomized, double-blind, placebo-controlled phase 2b trial over 12 weeks in patients with at least moderately active CSU and an inadequately controlled disease. The study showed a statistically significant dose response measured as a change in the urticaria activity score from baseline (UAS7) compared to placebo at the fourth week [135]. Fenibrutinib was also tested in CSU and negative results were reported [9].

### 13.2. IgG4-Related Disease

An IgG4-related disease is a newly recognized fibroinflammatory disorder characterized by elevated IgG4 serum levels, increased infiltration of IgG4-positive plasma cells, and fibrosis in involved organs and swelling of organs and tissues [136]. Treatment is typically based around systemic glucocorticoids, immunosuppressive drugs, biological agents, and surgery [137]. However, despite improvements being seen in most patients, relapse remains common after IgG4-RD. BTK inhibitors can be potentially useful in treatment and phase 2 clinical trials with rilzabrutinib and zanubrutinib were recently initiated (Table 2) [138]. However, the results are not available yet.

### 13.3. Graft Versus Host Disease

Acute and chronic graft versus host disease is a life-threatening complication after allogenic hematopoietic cell transplantation (allo-HCT). Currently, approved strategies for GVHD treatment are immunosuppressive therapies, mainly glucocorticoids. However, although therapeutic options for glucocorticoid-refractory patients are limited, BTK inhibitors have shown promising results in preclinical animal models and clinical trials [139]. Treatment with ibrutinib improved acute GVHD clinical scores and led to a longer survival in mice models of acute GVHD [140].

A phase 1b/2 study was conducted to determine the safety and efficacy of ibrutinib in chronic GVHD patients who had failed at least one line of treatment (Table 1) [141]. The overall response (OR) rate was 76%, and 71% of patients showed a response for more than 20 weeks. At a median follow-up of 13.9 months, 29% of patients were still on ibrutinib treatment. The iNTEGRATE phase 3 clinical trial evaluated the combination with prednisone in patients with newly diagnosed, moderate to severe chronic GVHD after allo-HCT (Table 1). The ibrutinib arm yielded a higher response rate; in this group, corticosteroids could be withdrawn at 21 and 24 months and the patients demonstrated improved Lee symptom scores [142]. In 2017, ibrutinib was approved by the FDA for chronic GVHD after the failure of one or more lines of systemic therapy. Acalabrutinib is also under evaluation for GVHD prophylaxis following allogeneic SCT in a phase 2 study (Table 1).

## 14. Safety Profile

BTK inhibitors have a specific safety profile that requires monitoring and management. Indeed, the most common reason for treatment discontinuation is toxicity, with BTKi-specific adverse events (AEs) including atrial fibrillation (AF), bleeding events, arthralgias, rash, diarrhea, and cytopenias [6]. Acalabrutinib and zanubrutinib have less off-target activity than ibrutinib and are better tolerated. The most common AEs are headaches, observed in patients treated with acalabrutinib, and neutropenia and infections, in patients treated with zanubrutinib [6,143]. Patients treated with BTKis are immunocompromised and have a high risk of infections, which develop in more than 50% of patients [143]. Treatment with BTKis is also associated with multiple immune defects, including decreased immunoglobulin levels and neutrophil and macrophage dysfunction; however, the IgA level has been observed to increase in some patients. In a recent cross-trial comparison of BTK inhibitors, similar immunologic changes were observed with both ibrutinib and acalabrutinib; both were associated with a sustained increase in serum immunoglobulin A [144]. However, in a recent preclinical study, ibrutinib reduced the primary antibody responses against an Adeno-associated virus (AAV) capsid [145]. 

It is not clear whether BTKis affect the risk or severity of COVID-19 or reduce vaccine efficacy [146]. BTK regulates the activity of macrophages, and it has been suggested that BTKis reduce inflammatory symptoms in COVID-19 patients with a hyperinflammatory immune response [147]. Moreover, acalabrutinib induced a clinical improvement, and reduced inflammation in patients with severe COVID-19. However, it is important to note that BTK inhibitors impair the innate immunity and increase susceptibility to infections. Despite this, long-term BTKi therapy may improve the recovery of humoral immune function, decrease infection rates, and protect patients from lung injury in the event of COVID-19 infection [148]. BTKis are likely to decrease direct B-cell–mediated responses to COVID-19 vaccines, as they inhibit the proliferation, differentiation, and class switching essential for antibody production [149]. In a recent study performed in CLL patients, treatment with BTK inhibitors or IgA deficiency were independently associated with failure to generate an antibody response after the second vaccine [150]. 

Ibrutinib and other BTKis are not approved for the treatment of RA, SLE, MS, or other autoimmune diseases, mainly due to toxicities related to their off-target effects. However, the development of new, more specific drugs can change this situation.

## 15. Conclusions

BTK is a signaling kinase that plays a crucial role in the activation of pathogenic B cells and autoantibody production in human autoimmune disorders. It also plays an important role in the production of pathogenic cytokines. In the past few years, BTK has become a new target for the treatment of B-cell lymphoid malignancies. Ibrutinib, a first-generation BTKi, revolutionized the treatment of such diseases. The second-generation BTKis, such as acalabrutinib and Zanubrutinib, offer greater BTK selectivity. Third-generation agents, including the non-covalent BTK inhibitors pirtobrutinib and nemtabrutinib, have entered later-stage clinical development in lymphoid malignancies. 

More recently, BTK inhibitors have shown promise for treating autoimmune diseases and other immune disorders in preclinical studies and early clinical trials. The results of preclinical studies from autoimmune animal models are promising, and several phase 1 and phase 2 clinical trials with BTK inhibitors were initiated in patients with immune disorders, such as RA, SLE, SM, PV, AIHA, ITP, and GVHD. Among these, positive results have been obtained for MS, PV, ITP, AIHA, RA, and GVHD, while SLE and SD have not been so promising. It is not currently clear why these agents demonstrate lower efficacy against some immunologic diseases. It is possible that BTKi treatment may be associated with a low BTK occupancy in these disorders, or that the suppressive activity of BTK inhibition in autoreactive B lymphocytes can partly be reduced by the inhibitor-induced activation of autoreactive T cells. The new generation of irreversible and reversible BTKis are currently under investigation in immunological disorders; some of them, including acalabrutinib and zanubrutinib, have been found to be more specific than ibrutinib and induce less adverse events, including atrial fibrillation and hemorrhage.

Novel BTKis with a higher specificity are currently being developed and are under evaluation in clinical trials. In addition, the combination of BTKis with other agents may be more effective than monotherapy. Future research should examine the pathogenic role of BTK signaling and its inhibition in autoimmune and inflammatory disorders. In addition, combination strategies based on BTK inhibitors and more specific novel drugs may yield positive results in immunological diseases.

## Figures and Tables

**Table 1 jcm-11-02807-t001:** Irreversible BTK inhibitors studied in immune disorders.

BTK Inhibitor [References]	Characteristics	Clinical Trials
Ibrutinib (PCYC-1102, Imbruvica, Pharmacyclics/Janssen) [19]	Covalent BTKi with off-target activity (EGFR, ErbB2, ITK and TEC), IC50 = 0.5 nM	AIHA (Ph 2—NCT03827603, NCT04398459); GVHD (Ph 1—NCT02195869, Ph 2—NCT04961801, Ph3—NCT02959944)
Acalabrutinib (ACP-196, Calquence, Acerta Pharma, AstraZeneca) [20,21,22,23,24]	Covalent, highly selective, BTKi, IC50 = 5.1 nM	RA (Ph 2—NCT02387762), AIHA (Ph 2—NCT04657094), GVHD (Ph 2—NCT 04198922, NCT04716075).
Zanubrutinib (BGB-3111, Brukinsa, BeiGene) [25,26]	Highly selective, covalent BTKi, lower off-target inhibitory activity on ITK, JAK3, and EGFR. C50 = 0.5	ITP (Ph2—NCT05214391), AS (Ph 2—NCT05199909), IgG4-RD (Ph 2—NCT04602598), SLE (Ph2—NCT04643470).
Spebrutinib (CC-292, AVL-292, Avila Therapeutics/Celgene) [27,28]	Covalent, highly selective BTKi, near-complete BTK occupancy for 8–24 h. IC50 < 0.5 nM	RA (Ph 2—NCT01975610).
Evobrutinib (A18, M2951, Merck) [29,30]	Covalent, highly selective BTKi, both for BCR and Fc receptor signaling, IC50 = 9 nM	MS (Ph3—NCT04032171, NCT04032158, NCT04338022), RA (Ph2—NCT03233230), SLE (Ph2—NCT02975336)
Remibrutinib (LOU064, Novartis) [31,32,33]	Covalentl BTKi and TEC inhibitor in vitro, inhibits BTK-dependent platelet activation, IC50 = 1.3 nM	CSU (Ph3—NCT05030311), SD (Ph2—SLOUiSSe, NCT04035668)
Tirabrutinib (Velexbru^®^, ONO/GS-4059, Ono Pharmaceutical) [34,35,36,37]	Specific, covalent BTKi, IC50 = 2 nM	SD (Ph 2—NCT03100942), RA (Ph 1—NCT02626026).
Elsubrutinib (ABBV-105, Abbvie) [38]	Covalent, selective BTKi, inhibits histamine release from IgE-stimulated basophils and IL-6 release from IgG-stimulated monocytes, IC50 = 0.18 µM	RA (Ph 2—NCT03682705); SLE (Ph 2—NCT03978520, NCT04451772).
Tolebrutinib (SAR 442168, PRN 2246, Sanofi/Principia Biopharma) [39,40]	Covalent, BTKi with immunomodulatory activities, it can cross the blood–brain barrier, IC50 = 0.4–0.7 nM	MS (Ph 2—NCT04742400; Ph3—NCT04411641, PERSEUS, NCT04458051-, NCT04410991, GEMINI 1—NCT04410978, GEMINI 2, NCT04410991); MG (Ph 3—URSA NCT05132569)
Orelabrutinib (ICP-022, Biogen/Innocare Pharma) [41]	Covalent, selective BTKi, IC50 = 1.6 nM	ITP (Ph 2—NCT05020288, NCT05124028), SLE (Ph 1/2—NCT04305197), MS (Ph 2—NCT04711148)
Branebrutinib (BMS-986195, Bristol-Myers Squibb), [42,43]	Covalent BTKi, 5000-fold higher selectivity for BTK over 240 other kinases, IC50 = 0.1 nM	AD (Ph 2—NCT05014438), PS (Ph 2—NCT02931838), RA (Ph 1—NCT03245515, Ph 1—NCT03131973), SLE, SS (Ph 2—NCT04186871).
Poseltinib (HM71224, LY3337641, Hanmi Pharmaceutical, Eli Lilly) [44,45,46]	Selective, non-covalent BTKi with potential anti-inflammatory activity, IC50 = 1.95 nM	RA (Ph 2—NCT02628028)
SHR1459 (TG 1701, EBI-1459; Reistone Biopharma, Jiangsu Hengrui Medicine Co.) [47,48]	Covalent, selective BTKi, IC50 = 6.70 nM	NO (Ph 2—NCT04670770); MG (Ph 2—NCT05136456)
TAS5315 (SAT0056, Taiho Pharmaceutical Co.), [49,50]	Covalent, highly selective BTKi with Cys481, IC50 < 0.15 nM.	RA (Ph 2—NCT03605251)
AC0058 (ACEA Biosciences, Inc, Sorrento Therapeutics) [5]	Covalent BTKi, inhibits B-cell activation and inflammatory cytokine production in monocytes.	SLE (Ph 1—NCT03878303)

Abbreviations: AD—atopic dermatitis, AIHA—autoimmune hemolytic anemia, AS—antiphospholipid syndrome, BTKi—Brutton kinase inhibitor, CSU—chronic spontaneous urticaria, GVHD—graft versus host disease, IgG4-RD—IgG4-related disease, TP—immune thrombocytopenia, MG—myasthenia gravis, MS—multiple sclerosis, NO—neuromyelitis optica, Ph—phase, RA—rheumatoid arthritis, SLE—systemic lupus erythematosus, Ph—phase, SD -Sjögren’s disease, RA—rheumatoid arthritis, SLE—systemic lupus erythematosus.

**Table 2 jcm-11-02807-t002:** Reversible Bruton’s kinase inhibitors studied in immune disorders.

BTK Inhibitor [References]	Characteristics	Clinical Trials
Fenebrutinib (GDC-0853; Roche/Chugai Pharmaceutical) [59,60,61,62,63,64]	Non-covalent BTKi, inhibits IgE-mediated histamine release from mast cells, IC50 = 0.91	RA (Ph 2—NCT02983227); SLE (Ph 2—NCT02908100, NCT03407482)
Rilzabrutinib (PRN1008; Principia Biopharma/Sanofi) [65,66,67,68]	Covalent BTKi, high affinity and selectivity for the BTK, prolonged, reversible target occupancy, anti-inflammatory effects, IC50 = 1.3 nM.	ITP (Ph 2—NCT03395210; Ph 3—LUNA 3, NCT04562766); AIHA (Ph2—NCT05002777); IgG4-RD (Ph 2—NCT04520451); PV (Ph2—NCT03762265); Asthma (Ph2—NCT05104892); CSU (Ph2—NCT05107115) and AD (Ph2—NCT05107115).
Nemtabrutinib (MK1026, ARQ 531; ArQule, Inc.) [69,70]	Non-covalent BTKi, inhibits signaling downstream of PCLG2, activity on SRC, ERK and act. IC50 = 0.8 nM.	Asthma (Ph 2—NCT01370317)
PRN473 (SAR 444727; Principia/Sanofi [71,72]	Covalent and noncovalent, BTKi, inhibits the activation of the β_2_-integrin c-1 and subsequently neutrophil recruitment into inflamed tissue; localized application to the skin; IC50 = 2.1–13.0 nM	AD (PPh 2—NCT04992546)
BMS-986142 (Bristol-Myers Squibb) [63,73]	Covalent BTKi with reduced FcR-mediated cytokine production and BCR-induced cytokine production; IC50 = 0.5 nM	RA (Ph 2—NCT02638948)

Abbreviations: AD—atopic dermatitis, BTKi—Bruton’s kinase inhibitor, IgG4-RD—IgG4-related disease, ITP—immune thrombosytopenia, Ph—phase, PV—pemphigus vulgaris, RA—rheumatoid arthritis, SLE—systemic lupus erythematosus.

## Data Availability

Not applicable.

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
