# Peer review of "Bruton’s Kinase Inhibitors for the Treatment of Immunological Diseases: Current Status and Perspectives"

_jcm, 2022, doi:10.3390/jcm11102807_

Round 1
Reviewer 1 Report
I found this review to be overall well written and comprehensive in describing evidence of BTKi for treatment of immunological diseases. Table 1 and 2 are useful, but would recommend making the second column more consistent in both tables - including for each BTKi whether it is covalent or non-covalent, reversible or irreversible (either state for all or don't state as this is in the table description), relevant off target known kinase inhibition, and IC50 and remove vague statements such as "more selective than ibrutinib". Would also recommend avoiding term "second generation" and providing more descriptive and consistent data listed above.
Author Response
Reviewer 1
I found this review to be overall well written and comprehensive in describing evidence of BTKi for treatment of immunological diseases.
Table 1 and 2 are useful, but would recommend making the second column more consistent in both tables - including for each BTKi whether it is covalent or non-covalent, reversible or irreversible (either state for all or don't state as this is in the table description), relevant off target known kinase inhibition, and IC50 and remove vague statements such as "more selective than ibrutinib".
Response: We thank the reviewer for positive review of our paper. The second column in both tables has been re-edited as requested.
Would also recommend avoiding term "second generation" and providing more descriptive and consistent data listed above.
Response: Corrected as requested

Reviewer 2 Report
This is an extremely thorough review and I highly commend the authors for such a detailed manuscript.
The text is very detailed, but may be a little overwhelming. Therefore, would it be possible to summarize the diseases that each BTKi have had trials for (already described in the text) for Tables 1 and 2 also. This would be veyr useful for readers as a point of reference.
Secondly, the authors begin the manuscript talking about XLA as the basis of discovering BTKi. I think it would be important for the readers to understand (more generally) the currently safety profile of BTKi, especially in the context of acquired immunodeficiency/antibody deficiency. Therefore, I would suggest that the authors can consider adding perhaps a passage/paragraph about the overall safety proofile of BTKi in general (following description of each disease). With a special comment/focus on the effect on Ab production.
Lastly, not all immunologists would regard CSU as an allergic disease (although Type I has been described as due to "autoallergy"). I would suggest to move CSU out of the category of allergic condition and perhaps to "other immunological disease"
Minor
Typo Line 80 l"ypmphoid"
Line 207 "aactivity"
Author Response
Reviewer 2
This is an extremely thorough review and I highly commend the authors for such a detailed manuscript.
Response: We thank the reviewer for positive review of our paper
The text is very detailed, but may be a little overwhelming. Therefore, would it be possible to summarize the diseases that each BTKi have had trials for (already described in the text) for Tables 1 and 2 also. This would be very useful for readers as a point of reference.
Response: We mentioned in the text clinical trials presented in table 1 and table 2 as much as possible
Secondly, the authors begin the manuscript talking about XLA as the basis of discovering BTKi. I think it would be important for the readers to understand (more generally) the currently safety profile of BTKi, especially in the context of acquired immunodeficiency/antibody deficiency. Therefore, I would suggest that the authors can consider adding perhaps a passage/paragraph about the overall safety proofile of BTKi in general (following description of each disease). With a special comment/focus on the effect on Ab production.
Response: We have added subchapter 13 about the safety profile of BTKi wiith a comment on the effect on Ab production, if data were available.
- 13. Safety profile
BTK inhibitors have a specific safety profile that requires monitoring and management. Indeed, the most common reason for treatment discontinuation is toxicity, with BTKi-specific adverse events (AEs) including atrial fibrillation (AF), bleeding events, arthralgias, rash, diarrhea, and cytopenias [6]. Acalabrutinib and zanubrutinib have less off-target activity than ibrutinib and are better tolerated. The most common AEs are headache, observed in patients treated with acalabrutinib, and neutropenia and infections, in patients treated with zanubrutinib [6,154]. Patients treated with BTKis are immunocompromised and have high risk of infections, which develop in more than 50% of patients [154]. Treatment with BTKis is also associated with multiple immune defects, including decreased immunoglobulin levels and neutrophil and macrophage dysfunction; however, IgA level has been observed to increase in some patients. In a recent cross-trial comparison of BTK inhibitors, similar immunologic changes were observed with both ibrutinib and acalabrutinib; both were associated with a sustained increase in serum immunoglobulin A [155]. However, in a recent preclinical study, ibrutinib reduced primary antibody responses against Adeno-associated virus (AAV) capsid [156].
It is not clear whether BTKis affect the risk or severity of COVID-19 or reduce vaccine efficacy [157]. BTK regulates the activity of macrophages, and it has been suggested that BTKis reduce inflammatory symptoms in COVID-19 patients with hyperinflammatory immune response [158]. Moreover, acalabrutinib induced clinical improvement, and reduced inflammation, in patients with severe COVID-19. However, it is important to note that BTK inhibitors impair the innate immunity and increase susceptibility to infections. Despite this, long-term BTKi therapy may improve recovery of humoral immune function, decrease infection rates and protect patients from lung injury in the event of COVID-19 infection [159]. BTKis are likely to decrease direct B cell–mediated responses to COVID-19 vaccines, as they inhibit the proliferation, differentiation, and class switching essential for antibody production [160]. In a recent study performed in CLL patients, treatment with BTK inhibitors or IgA deficiency were independently associated with failure to generate an antibody response after the second vaccine [161].
Ibrutinib and other BTKis are not approved for the treatment of RA, SLE, MS or other autoimmune diseases, mainly due to toxicities related to their off-target effects. However, the development of new, more specific drugs can change this situation.
Lastly, not all immunologists would regard CSU as an allergic disease (although Type I has been described as due to "autoallergy"). I would suggest to move CSU out of the category of allergic condition and perhaps to "other immunological disease"Response: CSU has been moved to "other immunological disease"
Minor Typo
Line 80 l"ypmphoid"
Line 207 "aactivity"
Response: Corrected as requested
